

# Predicting continuous ground reaction forces from accelerometers during uphill and downhill running: a recurrent neural network solution

Ryan S. Alcantara[1,4], W. Brent Edwards[2], Guillaume Y. Millet[3] and Alena M. Grabowski[1]

[1] Department of Integrative Physiology, University of Colorado Boulder, Boulder, CO, United States of America
[2] Human Performance Laboratory, Faculty of Kinesiology, University of Calgary, Calgary, Alberta, Canada
[3] Laboratoire Interuniversitaire de Biologie de la Motricité, Université Lyon, UJM-Saint-Etienne, Saint-Etienne, France
[4] Current affiliation: Department of Bioengineering, Stanford University, Stanford, CA, United States of America

## ABSTRACT

**Background**. Ground reaction forces (GRFs) are important for understanding human movement, but their measurement is generally limited to a laboratory environment. Previous studies have used neural networks to predict GRF waveforms during running from wearable device data, but these predictions are limited to the stance phase of level-ground running. A method of predicting the normal (perpendicular to running surface) GRF waveform using wearable devices across a range of running speeds and slopes could allow researchers and clinicians to predict kinetic and kinematic variables outside the laboratory environment.

**Purpose**. We sought to develop a recurrent neural network capable of predicting continuous normal (perpendicular to surface) GRFs across a range of running speeds and slopes from accelerometer data.

**Methods**. Nineteen subjects ran on a force-measuring treadmill at five slopes (0°, ±5°, ±10°) and three speeds (2.5, 3.33, 4.17 m/s) per slope with sacral- and shoe-mounted accelerometers. We then trained a recurrent neural network to predict normal GRF waveforms frame-by-frame. The predicted versus measured GRF waveforms had an average ± SD RMSE of 0.16 ± 0.04 BW and relative RMSE of 6.4 ± 1.5% across all conditions and subjects.

**Results**. The recurrent neural network predicted continuous normal GRF waveforms across a range of running speeds and slopes with greater accuracy than neural networks implemented in previous studies. This approach may facilitate predictions of biomechanical variables outside the laboratory in near real-time and improves the accuracy of quantifying and monitoring external forces experienced by the body when running.

Corresponding author
Ryan S. Alcantara,
ryan.alcantara@colorado.edu

## INTRODUCTION

Ground reaction forces (GRFs) are applied to the body when the foot is in contact with the ground and their measurement has facilitated numerous insights into the etiology of running-related injuries (*Ceyssens et al., 2019*). However, measurement of GRFs is generally restricted to a laboratory environment. To determine the effects of sport-specific environments on running kinetics and kinematics, previous studies have replicated aspects of an athlete's competitive environment (*e.g.*, running surface, slope) within a laboratory environment (*Voloshina & Ferris, 2015*; *Kipp, Taboga & Kram, 2017*; *Whiting et al., 2020*). Alternatively, inertial measurement units (IMUs; wireless wearable devices that measure magnetism, linear acceleration, and angular velocity) have been used to measure athletes' leg joint angles, stride kinematics, and segmental accelerations during competitive events (*Reenalda et al., 2016*; *Ruder et al., 2019*; *Clermont et al., 2019*). Although IMUs cannot directly measure GRFs, previous studies have used algorithms to estimate discrete biomechanical variables like peak vertical GRF, ground contact time, vertical impulse, and vertical loading rate (*Neugebauer, Hawkins & Beckett, 2012*; *Kiernan et al., 2018*; *Ancillao et al., 2018*; *Derie et al., 2020*; *Alcantara et al., 2021*) from IMU data.

Recently, neural networks have been used to predict GRF waveforms during running (*Wouda et al., 2018*; *Pogson et al., 2020*; *Dorschky et al., 2020*; *Johnson et al., 2021*), from which a variety of discrete variables can be calculated. Although predictions of the entire GRF waveform represent a more versatile outcome compared to predicting a discrete variable, previous studies have used neural network architectures that required waveforms to be normalized to the duration of a step (*Dorschky et al., 2020*) or stance phase (*Wouda et al., 2018*; *Johnson et al., 2021*). Temporal normalization is typically accomplished by identifying gait events in the GRF waveform and segmenting the neural network's input signals, preventing the calculation of biomechanical variables with a temporal component (*e.g.*, ground contact time, step frequency, vertical impulse, and loading rate) outside the laboratory where the GRF waveform is unavailable. Additionally, previous studies have predicted GRF waveforms only during level-ground running (*Wouda et al., 2018*; *Pogson et al., 2020*; *Dorschky et al., 2020*; *Johnson et al., 2021*), limiting the application to environments that are flat (*e.g.*, level treadmill or athletics track). Road and trail running are internationally popular forms of physical activity (*Running USA, 2019*; *International Trail Running Association, 2020*) and require runners to navigate a variety of uphill and downhill slopes. A method that accurately predicts GRF waveforms from wearable device data across a range of running slopes, while maintaining the temporal component, could allow researchers, clinicians, and coaches to measure and monitor a variety of kinetic and kinematic variables in outdoor environments.

Long short-term memory (LSTM) networks (*Hochreiter & Schmidhuber, 1997*) are a type of recurrent neural network that can overcome the traditional requirement of normalizing GRF waveforms to the duration of stance phase because LSTM networks can recurrently predict smaller, uniform portions of a larger sequence of any length. As such, a sequence of continuous GRF data can be predicted if it can be broken up into uniform portions. For the prediction of a given portion, LSTM networks use information from previous
portions, effectively "remembering" the portion's context, and have been used to predict sequential data during natural language processing tasks (*Wang & Jiang, 2016*). In the field of Biomechanics, LSTM networks have been used to make frame-by-frame predictions of GRF waveforms using motion capture data (*Mundt et al., 2020*) and predictions of the center of mass position relative to center of pressure from IMU data during walking (*Choi, Jung & Mun, 2019*). Developing an LSTM network to predict GRF waveforms from wearable device data would allow researchers to predict GRF waveforms not only during an isolated stance phase, but continuously for multiple steps or the entire duration of a run. IMUs have already been used to longitudinally measure biomechanical variables (*Reenalda et al., 2016*; *Kiernan et al., 2018*; *Ruder et al., 2019*; *Clermont et al., 2019*) and applying an LSTM network to such data could effectively provide a way to indirectly measure continuous GRF waveforms outside of the laboratory at a scale that was previously unattainable. Improving the accuracy of remotely measuring biomechanical variables with wearable devices may improve a clinician's ability to identify injury risk factors or, monitor rehabilitation progress (*Gurchiek, Cheney & McGinnis, 2019*).

The purpose of this exploratory study was to develop an LSTM network that could predict continuous normal (perpendicular to running surface) GRF waveforms across a range of running speeds and slopes using data from accelerometers. We sought to develop a network that could predict GRF waveforms with accuracy better than state-of-the-art predictions of time-normalized vertical GRF data during level-ground running using data from multiple IMUs: a root mean square error (RMSE) of 0.21 BW (*Dorschky et al., 2020*) and relative RMSE (rRMSE; RMSE normalized to the average range of the compared waveforms; Eq. 1) of 13.92% (*Johnson et al., 2021*).

## MATERIALS & METHODS

### Subjects

We analyzed a pre-existing dataset (*Baggaley et al., 2019*; *Khassetarash et al., 2020*; *Vernillo et al., 2020*) where 21 subjects ran at a combination of running speeds and slopes. Two subjects were excluded from the current analysis due to equipment data acquisition errors, leaving 19 subjects remaining (10 male, nine female; 29 ± 9 years, 173 ± 9 cm, 68.1 ± 9.9 kg). All subjects provided informed written consent and the experimental protocol was approved by the University of Calgary Conjoint Health Research Ethics Board (#REB14-1117).

### Experimental protocol

Following a 5 min warm up at a self-selected speed, each subject completed thirty 30 s trials on a force-measuring treadmill (2,000 Hz; Bertec, OH, USA), which included five slopes (0°, ±5°, ±10°) at three speeds (2.5, 3.33, 4.17 m/s) per slope, and three step frequencies (preferred and ±10%) at 3.33 m/s for each slope. Three custom biaxial accelerometers (2,000 Hz) were adhered with tape to subjects during all conditions: two on the right shoe and one on the sacrum. The accelerometers on the shoe were only used to determine the foot strike pattern for each condition using a previously validated method (*Giandolini et al., 2014*), which provided the percentage of a trial's foot strikes classified as either rearfoot,

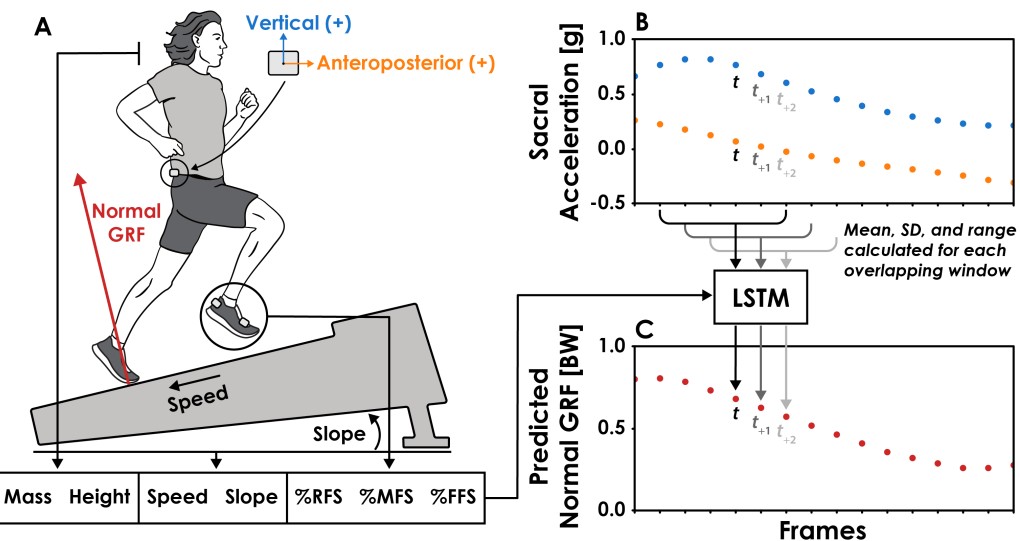

**Figure 1** **Overview of the long short-term memory (LSTM) network's input features and function.** (A) The LSTM network's input features included body mass, height, running speed, slope, and percentage of a trial's steps classified as rearfoot (RFS), midfoot (MFS), or forefoot (FFS) strikes. (B) Vertical and anteroposterior sacral acceleration data were divided into overlapping six frame (12 ms) windows, one for each frame of the normal ground reaction force (GRF) data. The mean, standard deviation (SD), and range of vertical and anteroposterior sacral acceleration values were calculated for each window and used as input features to the LSTM network. For the prediction of a normal GRF value at a given time ($t$), the respective window of acceleration data begins at $t_{-3}$ and ends at $t_{+2}$. (C) Normal GRFs were predicted frame-by-frame by the LSTM network using the 13 input features.

midfoot, or forefoot strikes. The biaxial accelerometer was placed on the sacrum such that the vertical axis in the accelerometer's local coordinate system was oriented superiorly, but we did not perform a calibration to align accelerometer and treadmill coordinate systems (Fig. 1). We did not align the accelerometer and treadmill coordinate systems because an LSTM network can likely learn the transformation between the coordinate systems and requiring this preprocessing calibration may limit the utility of an LSTM network outside a laboratory setting.

## Data processing

We analyzed 5 s of data (approximately 13 foot-ground contacts) from each trial and downsampled the normal GRF, vertical sacral acceleration, and anteroposterior sacral acceleration to 500 Hz to reduce the computational cost and match the sampling frequency of prior studies (*Day et al., 2021*; *Alcantara et al., 2021*). We normalized GRFs to bodyweight (BW) and filtered them using a 4th order low-pass Butterworth filter with a 30 Hz cut-off. We filtered the sacral acceleration data with a 4th order low-pass Butterworth filter with a 20 Hz cut-off. Preliminary analysis revealed that a 20 Hz cut-off improved prediction accuracy compared to 5 Hz, 10 Hz, 30 Hz, and no filter and preserved approximately 89% and 82% of the vertical and anteroposterior signal power, respectively.

Vertical sacral accelerometer data were further processed so that all negative values were replaced with zeros. Vertical center of mass acceleration is primarily negative during the

aerial phase of running (*Blickhan, 1989*) and preliminary analysis revealed that replacing negative vertical sacral accelerometer data with zeros helped the LSTM network avoid predictions of negative normal GRFs during the aerial phase. For each condition, we used the 2,500 frame (5 s trial @ 500 Hz) sequences of vertical and anteroposterior sacral accelerometer data to predict the simultaneously collected 2500 frame sequence of normal GRFs. The recurrent nature of the LSTM network requires sequential data to be divided into smaller portions that are iteratively used to make predictions. To accomplish this, we divided acceleration data for each trial into overlapping windows with a 6 frame (12 ms) width and padded the beginning and end of each trial's acceleration data with the first and final values, respectively, to ensure the number of windows equaled the number of normal GRF frames (2500) and that the windows were centered on the corresponding frame of the normal GRFs (Fig. 1). Preliminary analysis revealed that a window width of 6 frames was the smallest window we could use without decreasing LSTM network prediction accuracy and we found no improvement in prediction accuracy when using window sizes up to 60 frames (120 ms). Thus, the LSTM network iteratively predicted a single frame of the normal GRF at time $t$ using acceleration data from frames $t_{-3}$ through $t_{+2}$ (Fig. 1).

## Feature engineering

A total of 13 features were used as inputs in the LSTM network (Fig. 1). We calculated the mean, standard deviation (SD), and range of vertical and anteroposterior acceleration data for each 12 ms window and used them as input features. The use of summary statistics as input features has been shown to maintain neural network accuracy while benefiting from a reduced computational cost (*Figo et al., 2010*). These three summary statistics were normalized to a range of 0–1 and represent 6 (3 features × 2 acceleration axes) of the 13 input features. The remaining input features were selected due to their effect on running kinetics and kinematics: subject height, body mass, running speed, slope, and percentage of steps classified as either rearfoot, midfoot, or forefoot strikes (*Almeida, Davis & Lopes, 2015*; *Khassetarash et al., 2020*; *Vernillo et al., 2020*; *Vincent et al., 2020*; *Alcantara et al., 2021*). We chose not to include step frequency as an input feature, despite the presence of the ±10% preferred step frequency conditions, to increase the variability in the data used to predict GRF waveforms. Doing so theoretically represents a greater challenge for the LSTM network as there is additional variability between trials that is not being explicitly accounted for with an input variable.

## Neural network architecture

The neural network consisted of a Bidirectional LSTM and a multilayer perceptron (MLP) with three fully connected layers containing 128, 384, and 320 neurons, respectively (Fig. 2). The Bidirectional LSTM consists of two LSTM layers where the order of the input sequence is reversed for the second layer. Reversing the sequence for the second LSTM layer allows the network to utilize information from future portions of the sequence just as the first LSTM layer utilizes information from prior portions. The outputs from each LSTM layer are then averaged before being passed along to the MLP. The number and size of the layers were determined using the Hyperband hyperparameter optimization algorithm

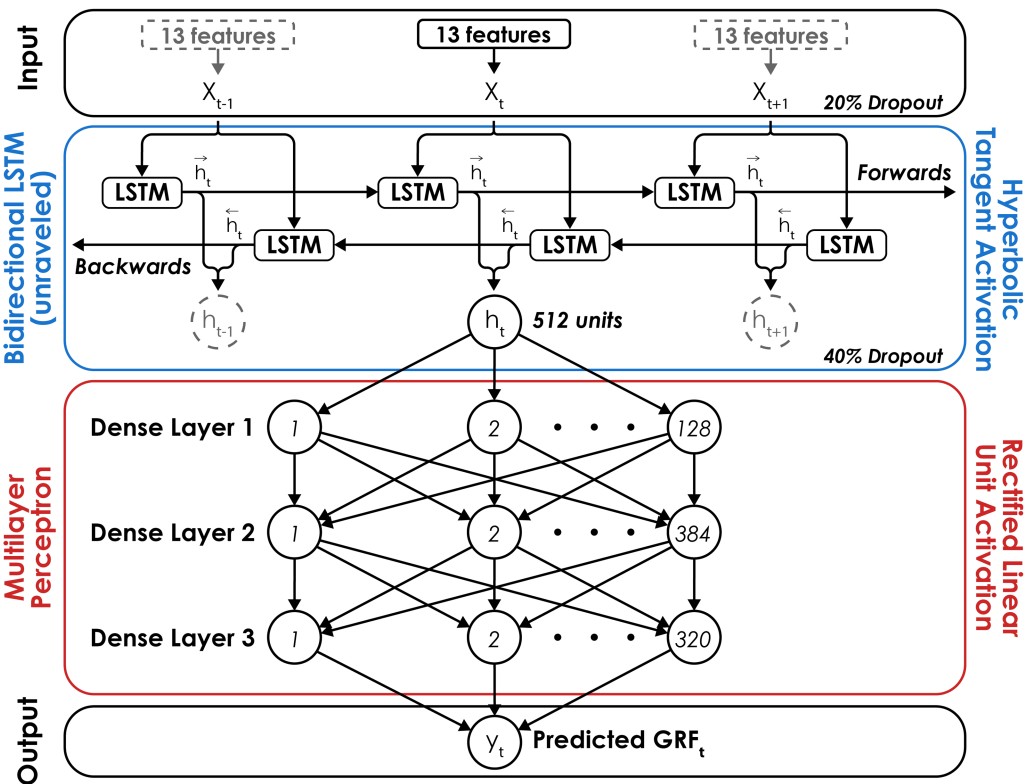

**Figure 2  Neural network architecture.** The long short-term Memory (LSTM) network consisted of a Bidirectional LSTM layer with a hyperbolic tangent activation function followed by a multilayer perceptron (MLP) with rectified linear unit activation functions for three hidden layers with 128, 384, and 320 neurons, respectively. The Bidirectional LSTM layer is unraveled to illustrate its recurrent nature and dashed lines signify inputs $(x)$ and outputs $(h)$ at time $t_{-1}$ and $t_{+1}$. A dropout rate of 20% was applied to the input layer of the network and a dropout rate of 40% was applied to the output of the Bidirectional LSTM layer to limit network overfitting. For each prediction of the normal ground reaction force (GRF) at a given time $(t)$, the network received 13 features as inputs $(x_t;$ Fig. 1), passed the output from the Bidirectional LSTM layer $(h_t)$ to the MLP, and predicted a single value $(y_t)$ with a linear activation function in the output layer.

(*Li et al., 2018*) on the data of two randomly selected subjects. The LSTM network was trained using a batch size of 32, learning rate of 0.001, and mean square error loss function. Network weights and biases were updated using the adaptive moment estimation (Adam) optimization algorithm at the end of each epoch (*Kingma & Ba, 2017*) and training lasted a maximum of 1,000 epochs or until the mean square error failed to decrease by 0.001 BW after 30 consecutive epochs. The neural network was developed using the Tensorflow (v2.2.0) python library (*Abadi et al., 2016*).

## Network validation

We assessed the accuracy and generalizability of the network using a Leave-One-Subject-Out (LOSO) cross validation method (*Halilaj et al., 2018*). LOSO cross validation is a variation of K-fold cross validation that requires the dataset to be subset by subject, with one subject's data withheld for testing purposes and the rest of the subjects' data used

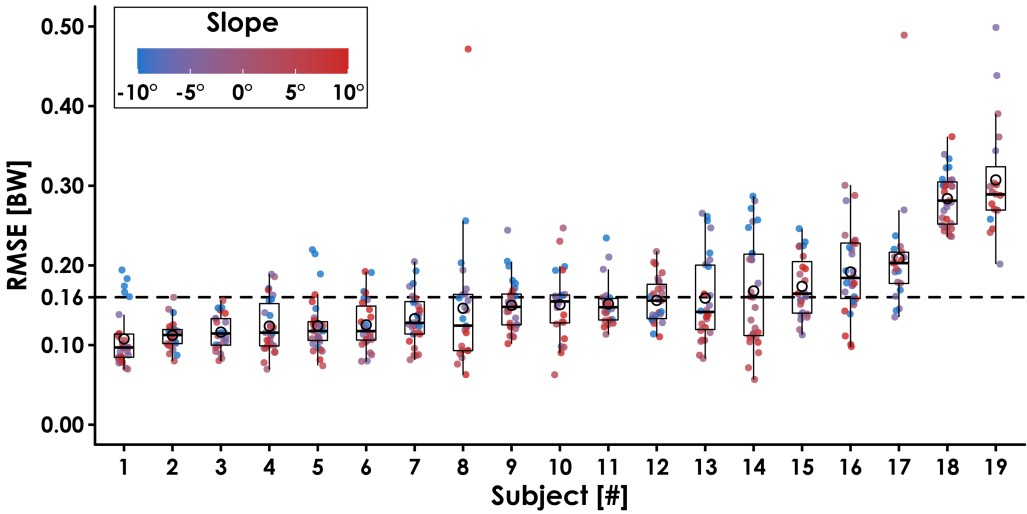

**Figure 3** **Ground reaction force waveform prediction error for each subject across all conditions.** The average root mean square error (RMSE) across subjects was 0.16 BW (dotted line). Filled circles represent each trial, and the color indicates slope (0°, ±5°, ±10°) at three speeds (2.5, 3.33, 4.17 m/s). Open circles represent each subject's average RMSE, horizontal bars are the median RMSE, box plot edges indicate the interquartile range (IQR; 25th and 75th percentile), and the whiskers encompass values that fall within 1.5*IQR. Subjects are sorted from lowest to highest RMSE.

to train the network. This process is repeated until the network has been tested on every subject's data, ultimately providing an ensemble of networks and their respective accuracy metrics. Performing LOSO cross validation can be computationally costly, as the network must be trained and tested a number of times equal to the number of subjects ($n = 19$), but the benefits of this method include the ensemble of accuracy metrics and assurance that a given subject's data are not included in both the training and testing subsets, which can artificially increase the reported accuracy of a network (*Saeb et al., 2017*; *Chaibub Neto et al., 2019*).

In addition to the LOSO cross validation method, we performed a test-train split of one representative subject's data according to slope (±5° trials reserved for testing, 0° and ±10° slopes used for training) to test the accuracy of the model when predicting speed-slope combinations that were not present during training. We selected Subject 14 as a representative subject because their RMSE during LOSO cross validation was similar to the average RMSE across all subjects (Fig. 3) and their GRF waveforms illustrated an interaction between running slope and normal GRF impact peak magnitude (*Gottschall & Kram, 2005*). This single-subject validation method prioritizes accuracy over generalizability (ability to make accurate predictions for a variety of individuals) and represents a potential circumstance where an LSTM network is trained on data collected from a single athlete prior to their competitive season and later used to predict only that athlete's GRF data from wearable device data during their competitive season.

Prediction error for each trial's GRF waveforms was quantified as the root mean square error (RMSE) and relative RMSE (rRMSE), which is RMSE normalized to the average
range of compared waveforms, expressed as a percentage and defined as

$$rRMSE = \frac{RMSE}{0.5 \times \sum_{i=1}^{2}(\max(x_i) - \min(x_i))} \times 100, \qquad (1)$$

where $x_1$ and $x_2$ are the GRF waveforms predicted by the LSTM network and measured by the force-measuring treadmill (*Ren, Jones & Howard, 2008*). Additionally, we used a threshold of 5% BW to identify stance phase and calculated the active peak of the normal GRF waveform, normal impulse, normal GRF loading rate, contact time, and step frequency from the predicted and measured GRF data. The normal GRF active peak was calculated as the maximum normal GRF value occurring between 40–60% of stance phase because the magnitude of the impact peak can exceed the active peak during downhill running and occurs during early stance phase (0–30%) (*Gottschall & Kram, 2005*; *Vernillo et al., 2020*). We calculated impulse as the integral of the normal GRF waveform during the stance phase with respect to time, loading rate as the average slope of the normal GRF waveform during the first 25 ms of stance phase (*Yong et al., 2018*), contact time as the duration when the normal GRF was greater than 5% BW, and step frequency as the number of initial foot-ground contacts per second. We report the mean absolute percent error (MAPE) of these discrete variables for each subject. Data analysis was performed in python (v3.6.9) and R (v4.0.4) using custom libraries (*Wickham, 2009*; *Wickham, 2019*; *Alcantara, 2019*; *Pandas Development Team, 2020*; *R Core Team, 2020*; *Wickham et al., 2020*; *Virtanen et al., 2020*; *Harris et al., 2020*).

We enforced two biomechanical boundaries upon the predicted GRF waveforms to ensure that data fell within established biomechanical limits and could be used to calculate the discrete biomechanical variables of interest. First, the predicted GRF waveform had to have an equal number of foot-ground contacts as the GRF waveform measured by the force-measuring treadmill, determined using the same 5% BW threshold. Second, the step frequency over the duration of the predicted GRF waveform had to be ≤ 4 Hz. We selected these criteria based on previous research of running biomechanics, as thresholds of 5% BW have been previously used to identify the stance phase for the calculation of kinetic or kinematic variables (*Day et al., 2021*; *Alcantara et al., 2021*) and during uphill and downhill running, step frequency is ≤ 4 Hz (*Cavagna et al., 1997*; *Snyder & Farley, 2011*). Trials that failed to meet either of these criteria were used to calculate the LSTM network's overall prediction failure rate and removed from subsequent analyses.

## RESULTS

We analyzed 529 trials for the present study. The predicted GRF waveforms for 32 trials (6%) failed to meet one or both criteria and were considered failed predictions. Specifically, we identified 22 trials (4%) that required a threshold greater than 5% BW to identify an equal number of steps between predicted and measured GRF waveforms and 10 trials (2%) that had a step frequency greater than 4 Hz. Thus, 94% of GRF waveforms predicted by the LSTM network fell within the imposed biomechanical boundaries.

Leave-One-Subject-Out cross validation revealed that the LSTM network predictions of each subject's normal GRF waveforms had an average ± SD RMSE of 0.16 ± 0.04

**Table 1  Mean ± SD root mean square error (RMSE) and relative RMSE (rRMSE) for normal GRF waveforms predicted by the LSTM network compared to the measured normal GRF waveforms for each subject.**

| Subject | RMSE [BW] | rRMSE [%] |
|---|---|---|
| 1 | 0.11 ± 0.04 | 4.0 ± 1.1 |
| 2 | 0.11 ± 0.02 | 4.1 ± 0.7 |
| 3 | 0.12 ± 0.02 | 4.4 ± 0.9 |
| 4 | 0.12 ± 0.03 | 4.6 ± 1.2 |
| 5 | 0.12 ± 0.04 | 4.7 ± 0.9 |
| 6 | 0.12 ± 0.03 | 5.2 ± 1.4 |
| 7 | 0.13 ± 0.03 | 5.1 ± 1.2 |
| 8 | 0.15 ± 0.08 | 5.3 ± 3.3 |
| 9 | 0.15 ± 0.03 | 5.4 ± 1.0 |
| 10 | 0.15 ± 0.04 | 5.2 ± 1.3 |
| 11 | 0.15 ± 0.03 | 5.8 ± 0.8 |
| 12 | 0.16 ± 0.03 | 6.1 ± 1.1 |
| 13 | 0.16 ± 0.06 | 5.7 ± 1.5 |
| 14 | 0.17 ± 0.07 | 6.9 ± 2.5 |
| 15 | 0.17 ± 0.04 | 7.9 ± 2.0 |
| 16 | 0.19 ± 0.05 | 6.7 ± 1.7 |
| 17 | 0.21 ± 0.07 | 7.2 ± 2.5 |
| 18 | 0.29 ± 0.03 | 14.0 ± 1.7 |
| 19 | 0.31 ± 0.07 | 13.7 ± 2.7 |
| Mean ± SD | 0.16 ± 0.04 | 6.4 ± 1.5% |

BW (Fig. 3) and rRMSE of 6.4 ± 1.5% compared to GRF waveforms measured by the force-measuring treadmill across all conditions (Table 1). RMSE values were generally lower during slow uphill running (2.5 m/s, +10°; 0.13 BW) compared to fast downhill running (4.17 m/s, −10°; 0.20 BW) (Fig. 4 and 5). The MAPE for step frequency was 0.1 ± 0.1%, contact time was 4.9 ± 4.0%, impulse was 6.4 ± 6.9%, normal GRF active peak was 8.5 ± 8.2%, and loading rate was 27.6 ± 36.1% (Table 2).

The prediction error for one representative subject's (Subject 14) normal GRF waveforms at ±5° during single-subject validation was lower than those resulting from the LOSO cross validation, with an average ± SD RMSE of 0.08 ± 0.02 BW and rRMSE of 3.3 ± 0.9%. The MAPE of step frequency (0.1 ± 0.1%), contact time (3.0 ± 2.3%), impulse (2.5 ± 1.9%), normal GRF active peak (2.7 ± 2.0%), and loading rate (17.6 ± 16.9%) calculated from predicted GRF waveforms were also generally lower than those resulting from LOSO cross validation.

## DISCUSSION

We developed a recurrent neural network capable of predicting continuous normal GRF waveforms across a range of running speeds (2.5–4.17 m/s), slopes (0°, ±5°, ±10°), and step frequencies (preferred, ±10%) from accelerometer data. Our findings indicate that an LSTM network with the runner's mass, height, running speed, slope, foot strike pattern,

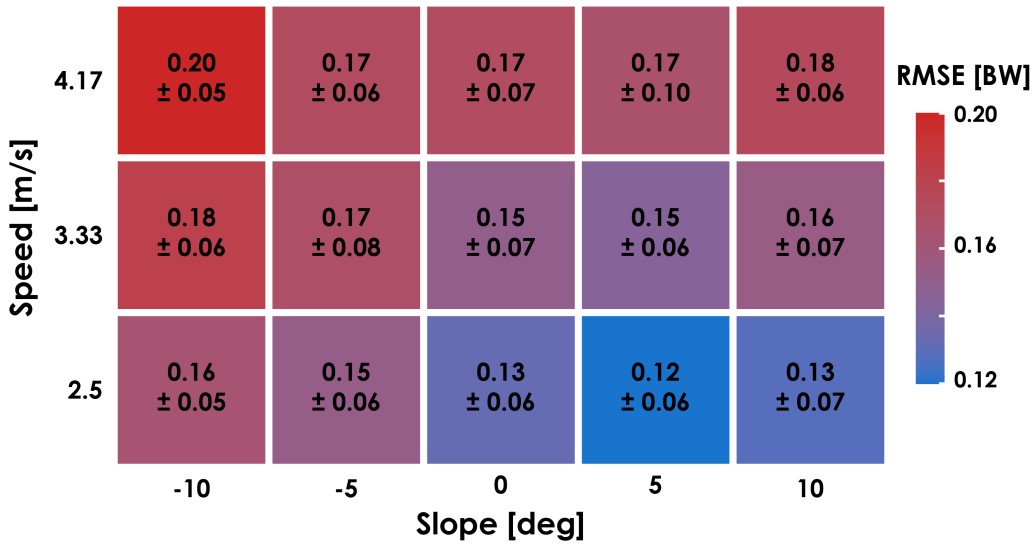

**Figure 4** **Ground reaction force waveform prediction error for each condition.** The average ± SD root mean square error (RMSE) of the predicted ground reaction force (GRF) waveforms compared to the GRF waveform measured by the force-measuring treadmill for each condition during leave-one-subject-out (LOSO) cross validation.

and sacral acceleration as input features can predict normal GRF waveforms across a range of speeds and slopes with an RMSE of 0.12–0.20 BW and rRMSE of 5.4–7.3% (Fig. 4). For comparison, previous studies report an RMSE of 0.39 ± 0.26 BW (*Wouda et al., 2018*), an RMSE of 0.21 ± 0.03 BW (*Dorschky et al., 2020*), and an rRMSE of 13.92% (*Johnson et al., 2021*) when using neural networks to predict the stance phase vertical GRF waveform during level-ground running. In contrast to previous studies, the LSTM network does not require preliminary stance phase identification or time normalization, which preserves the temporal component of the predicted GRF waveform. This characteristic of the LSTM network allowed us to calculate stride kinematic variables like step frequency and contact time with a MAPE < 5%. Additionally, the recurrent nature of the LSTM network facilitates frame-by-frame predictions of GRF waveforms and can be used to make predictions over any duration of running. Thus, an LSTM network could be used to quantify changes in normal GRF waveforms over the course of a prolonged run (*e.g.*, a marathon race).

The accuracy of predicted GRF waveforms varied across speeds and slopes, with a combination of faster running speeds and negative slopes producing greater RMSE values than slower running speeds and positive slopes (Fig. 4). The greater RMSE values during downhill running may be due to the LSTM network's inability to account for changes in impact peak magnitude across slopes (Fig. 5). Previous studies have found that the presence of an impact peak in the normal GRF waveform is subject-specific, affected by changes in running slope, and associated with acceleration of the effective mass of the lower extremity during early stance phase (*McMahon, Valiant & Frederick, 1987*; *Gottschall & Kram, 2005*; *Vernillo et al., 2020*). Thus, predictions of normal GRF waveforms across slopes may be further improved by incorporating accelerations measured at the feet or lower legs.

**Table 2  Mean ± SD of the discrete biomechanical variables.** Values were calculated from normal ground reaction force (GRF) waveforms predicted by the LSTM network ("Predicted") and normal GRF waveforms measured from the force-measuring treadmill ("Measured") across all speeds and subjects for each slope.

| Slope | Step frequency [Hz] | | Contact time [ms] | | Impulse [BW*s] | | Active peak [BW] | | Loading rate [BW/s] | |
|---|---|---|---|---|---|---|---|---|---|---|
| | Predicted | Measured | Predicted | Measured | Predicted | Measured | Predicted | Measured | Predicted | Measured |
| −10° | 3.1 ± 0.3 | 3.1 ± 0.3 | 215 ± 26 | 214 ± 29 | 0.33 ± 0.03 | 0.33 ± 0.05 | 2.38 ± 0.24 | 2.36 ± 0.40 | 64.7 ± 19.9 | 68.4 ± 20.6 |
| −5° | 3.1 ± 0.3 | 3.1 ± 0.3 | 222 ± 23 | 223 ± 27 | 0.34 ± 0.04 | 0.34 ± 0.04 | 2.41 ± 0.25 | 2.45 ± 0.36 | 54.9 ± 16.7 | 60.6 ± 19.4 |
| 0° | 3.1 ± 0.3 | 3.1 ± 0.3 | 228 ± 24 | 229 ± 27 | 0.33 ± 0.04 | 0.34 ± 0.04 | 2.42 ± 0.22 | 2.51 ± 0.36 | 42.0 ± 12.8 | 49.0 ± 17.2 |
| +5° | 3.2 ± 0.3 | 3.2 ± 0.3 | 228 ± 25 | 232 ± 26 | 0.32 ± 0.03 | 0.33 ± 0.04 | 2.36 ± 0.21 | 2.40 ± 0.33 | 34.8 ± 8.5 | 39.3 ± 13.3 |
| +10° | 3.4 ± 0.3 | 3.4 ± 0.3 | 223 ± 26 | 227 ± 26 | 0.30 ± 0.04 | 0.30 ± 0.04 | 2.24 ± 0.22 | 2.20 ± 0.29 | 30.1 ± 8.3 | 31.3 ± 12.7 |

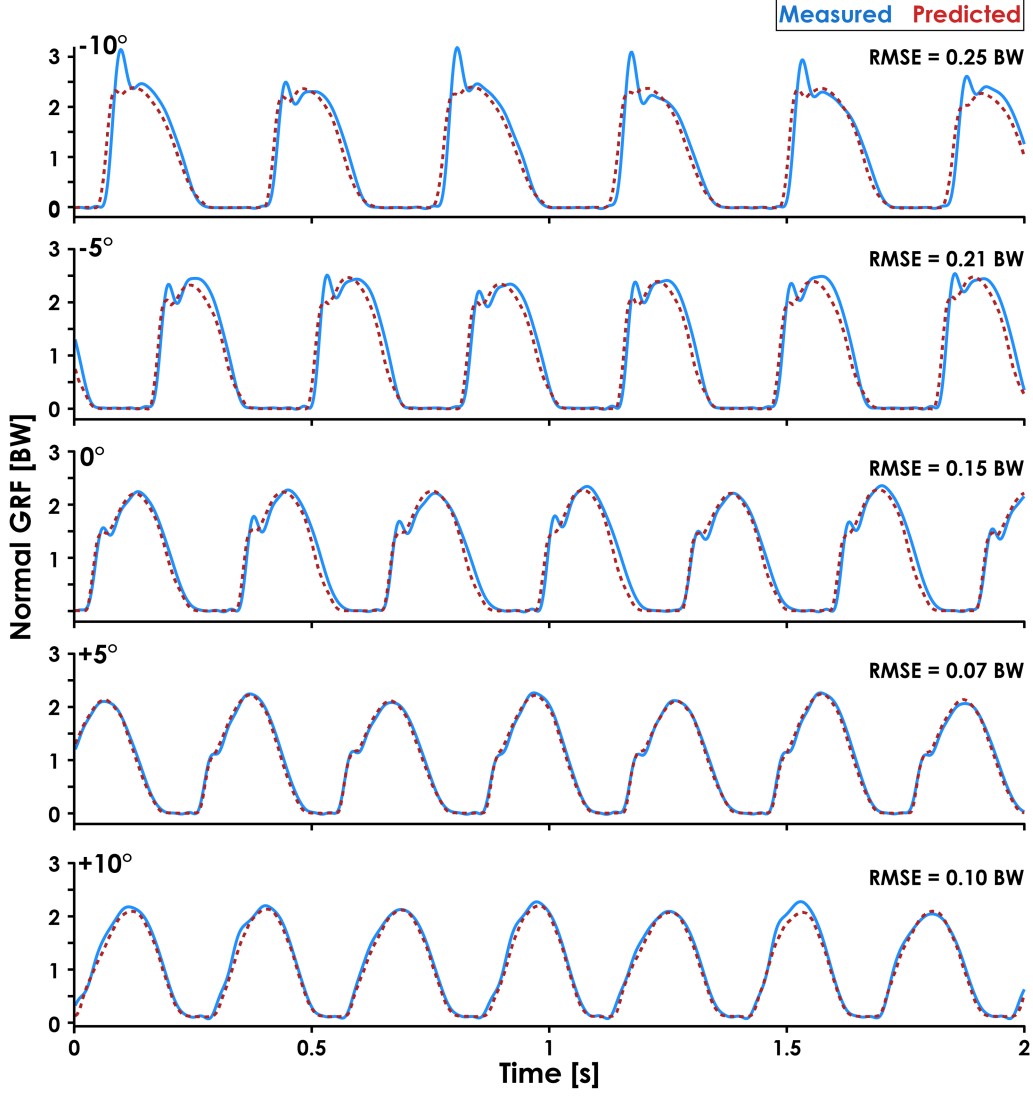

**Figure 5** **Predicted and measured normal GRF waveforms across slopes for a representative subject.** Normal ground reaction force (GRF) waveforms predicted by the recurrent neural network (dashed red lines) and measured by the force-measuring treadmill (solid blue lines) at 3.33 m/s and all slopes (0°, ±5°, ±10°) are presented for Subject 14. Subject 14 was selected because they had similar RMSE values (0.17 ± 0.07 BW) as the average across all subjects (0.16 ± 0.04 BW) and their GRF waveforms illustrate an interaction between running slope and normal GRF impact peak magnitude.

We also quantified the accuracy of the LSTM network when trained and tested on data from the same subject. Although not a valid method of determining the LSTM network's generalizability, single-subject validation provides insight into how well a personalized neural network could predict an individual's GRF waveforms for unknown combinations of speed and slope in the future. We found that predicted GRF waveforms of a representative subject (Subject 14) during the ±5° slope conditions had an average ± SD RMSE of 0.08 ± 0.02 BW, compared to 0.16 ± 0.03 BW during LOSO cross validation.

These findings indicate that a subject-specific LSTM network was twice as accurate as the LOSO cross validated LSTM network. A single-subject approach may be particularly beneficial for researchers, coaches, or clinicians who have the resources to train personalized LSTM networks and wish to monitor a specific athlete's biomechanics over the course of a competitive season. For example, an athlete could run at a variety of speeds and slopes while wearing accelerometers during a baseline data collection on a force-measuring treadmill at the start of their competitive season and a personalized LSTM network could be trained on their data. Then, if accelerometer data were collected from an athlete during training runs, their normal GRF waveforms and a variety of discrete values could be predicted and monitored longitudinally.

The MAPE values for step frequency, contact time, impulse, and normal GRF active peak were $\leq 8.5\%$, but the loading rate MAPE was $27.6 \pm 36.1\%$. The lower MAPE values for step frequency, contact time, impulse, and normal GRF active peak indicate that the LSTM network consistently identified the general shape of the GRF waveform and the boundaries of the stance phase despite changes in speed, slope, and step frequency. However, the network did not consistently predict the presence of an impact peak during early stance phase (Fig. 5, $-10°$ trial), which affected the predicted slope of the GRF waveform during early stance phase and thus the accuracy of loading rate values. Although the prominence of an impact peak in the normal GRF waveform is affected by foot strike pattern and slope (*Gottschall & Kram, 2005*), two of the inputs for the LSTM network, the decreased accuracy when estimating loading rate may be because we did not include accelerometer data from the lower extremities and impact accelerations are attenuated at the sacrum compared to the tibia (*Baggaley et al., 2019*). A previous study found moderate-strong correlations between axial tibial acceleration and vertical GRF impact peak magnitude ($r = 0.76$) and timing ($r = 0.94$) during running (*Hennig & Lafortune, 1991*), and future research aimed at improving the accuracy of loading rate estimates should include tibial acceleration as an input feature. We did not include accelerometer data from the shoes as inputs for the LSTM network because the data were not available for both feet.

Recurrent neural networks represent a promising strategy for predicting continuous normal GRFs from wearable devices in outdoor environments. The LSTM network required data from three accelerometers (one on the sacrum and two on the right shoe to determine foot strike pattern), but we also performed a *post-hoc* analysis of prediction accuracy without the foot strike pattern data to quantify the network's accuracy when only using data from one sacral accelerometer. The *post-hoc* analysis revealed that excluding foot strike pattern data slightly increased the average $\pm$ SD RMSE from $0.16 \pm 0.04$ BW to $0.17 \pm 0.05$ BW and rRMSE from $6.4 \pm 1.5\%$ to $6.7 \pm 1.7\%$. Excluding foot strike pattern data affected the MAPE of discrete variables by $< 3\%$ (Table 3). These findings indicate that the LSTM network can predict normal GRF waveforms from a single accelerometer on the sacrum more accurately than neural networks implemented in previous studies (RMSE = 0.21–0.39 BW, rRMSE = 13.92%), which required data from 3–7 wearable devices (*Wouda et al., 2018*; *Dorschky et al., 2020*; *Johnson et al., 2021*).

We further analyzed the importance of input features to the LSTM network by calculating prediction accuracy after systematically permuting each input feature across the trials for
**Table 3  Error metrics for the predicted waveforms and discrete variables when training the long short-term memory (LSTM) network with and without foot strike pattern as an input feature.** Root mean square error (RMSE) and relative RMSE (rRMSE) are presented for predicted normal ground reaction force (GRF) waveforms. Mean absolute percent error (MAPE) values are presented for the discrete variables calculated from normal GRF waveforms predicted by both LSTM networks.

|  | LSTM with foot strike (Sacral + Right foot accelerometers) | LSTM without foot strike (Only sacral accelerometer) |
|---|---|---|
| **GRF waveform** |  |  |
| RMSE [BW] | 0.16 ± 0.04 | 0.17 ± 0.05 |
| rRMSE | 6.4 ± 1.5% | 6.7 ± 1.7% |
| **Discrete variables MAPE** |  |  |
| Step frequency | 0.1 ± 0.1% | 0.1 ± 0.1% |
| Contact time | 4.9 ± 4.0% | 5.6 ± 4.5% |
| Impulse | 6.4 ± 6.9% | 6.0 ± 7.1% |
| Active peak | 8.5 ± 8.2% | 7.7 ± 6.3% |
| Loading rate | 27.6 ± 36.1% | 30.3 ± 41.6% |

the representative subject. This process of calculating Permutation Feature Importance (PFI) effectively severs the learned relationship between an input feature for a given trial and the corresponding GRF waveform (*Molnar, 2019*). PFI is calculated as the ratio between the RMSE of the LSTM network with a given feature permuted and the original prediction RMSE of the LSTM network. After 100 permutations for each input feature, we found that the input feature with the highest PFI was vertical acceleration (6.29), followed by anteroposterior acceleration (1.62), foot strike pattern (1.14), slope, (1.13), speed (1.07), height (1.00), and body mass (1.00). These findings indicate that the inclusion of body mass and height did not improve prediction accuracy and that the LSTM network relies most on the acceleration data when predicting the normal GRF waveform across a range of speeds and slopes.

Using a recurrent neural network in combination with accelerometers and a global positioning system (GPS) device to obtain speed and slope data could potentially allow runners to receive biomechanical feedback during an outdoor run. Watches with GPS capabilities are commonly used by runners (*Janssen et al., 2020*), have been used to provide real-time feedback of step frequency (*Willy et al., 2016*), and could provide running speed and slope data to the LSTM network to predict GRF waveforms in near-real time (*Scott, Scott & Kelly, 2016*). Discrete biomechanical variables could then be calculated from predicted normal GRF waveforms and sent to a clinician, coach, researcher, or the runner themselves. A similar approach has been implemented during outdoor walking and running using an integrated IMU-GPS device placed in a backpack, but it is unclear how accurate or generalizable this approach is as the network was trained and tested on data from three subjects and the reported accuracy metrics were combined for walking and running (*Davidson et al., 2019*). To facilitate calculation of GRF-based variables during outdoor running using accelerometers, we have made the LSTM networks, which were trained on all subjects, with and without the need for foot strike data, publicly available at

http://www.github.com/alcantarar/Recurrent_GRF_Prediction. We have included a tutorial on how to use an LSTM to continuously predict a signal from wearable device data, an approach that may be used to improve a clinician's ability to remotely quantify a patient's GRFs or monitor rehabilitation progress (*Gurchiek, Cheney & McGinnis, 2019*).

There are potential limitations to consider alongside our findings. The accelerometers used in the present study were biaxial and the inclusion of mediolateral sacrum accelerations may have further improved prediction as the mediolateral behavior of the center of mass is sensitive to changes in running speed (*Nilsson & Thortensson, 1989*) and slope. Additionally, accelerometers were adhered to subjects using tape and a less secure attachment method may introduce movement artefact into the accelerometer data. Previous research suggests that attachment method can affect peak tibial acceleration during running (*Johnson et al., 2020*), but the lower leg experiences larger accelerations than the sacrum during running (*Baggaley et al., 2019*) and thus is more sensitive to different attachment methods. However, variations in accelerometer orientation between subjects may have contributed to the range of RMSE values (0.11–0.31 BW) during LOSO cross validation (*Tan et al., 2019*). Additionally, variations in soft tissue movement artefact between the subjects used to train the LSTM network and other populations may introduce prediction error (*Peters et al., 2010*). Using the LSTM network to predict normal GRF waveforms from a sacral accelerometer adhered differently than in the present study may affect prediction accuracy, but the 20 Hz low-pass filter we applied to the accelerometer data can potentially mitigate this effect. Additionally, predictions made with the LSTM network may not be generalizable for speeds or slopes that fall outside the range of the training data (2.5–4.17 m/s and ±10°), for different running surfaces, as biomechanics change when running on steep slopes (*e.g.*, 20–40°) (*Giovanelli et al., 2016*; *Whiting et al., 2020*), for prolonged runs, on variable terrain (*Voloshina & Ferris, 2015*), with changes in speed (*Alcantara et al., 2021*), or in response to muscle fatigue (*Derrick, Dereu & McLean, 2002*). Lastly, the LSTM network was trained on data collected on a stiff force-measuring treadmill and thus accelerometer data collected during running on less stiff surfaces (*e.g.*, grass) may result in greater prediction errors given the effects of surface stiffness on running biomechanics and thus energy absorption (*Derrick, Hamill & Caldwell, 1998*; *Ferris, Louie & Farley, 1998*).

## CONCLUSIONS

We developed a recurrent neural network that used accelerometer data to predict continuous normal GRF waveforms across a range of running speeds (2.5–4.17 m/s) and slopes (0°, ±5°, ±10°) with an average ± SD RMSE of 0.16 ± 0.04 BW and rRMSE of 6.4 ± 1.5%. Unlike neural networks implemented in prior studies, the recurrent neural network does not require preliminary identification of the stance phase or temporal normalization and allows for near real-time predictions of normal GRF waveforms during running. Accurate predictions of normal GRF waveforms using wearable devices will

improve the ability to longitudinally monitor biomechanical variables in non-laboratory environments.

### Funding

We utilized resources from the University of Colorado Boulder Research Computing Group, which is supported by the National Science Foundation (awards ACI-1532235 and ACI-1532236), the University of Colorado Boulder, and Colorado State University. The funders had no role in study design, data collection and analysis, decision to publish, or preparation of the manuscript.

### Grant Disclosures

The following grant information was disclosed by the authors:
The National Science Foundation: ACI-1532235, ACI-1532236.

### Competing Interests

The authors declare there are no competing interests.

### Author Contributions

- Ryan S. Alcantara analyzed the data, prepared figures and/or tables, authored or reviewed drafts of the paper, and approved the final draft.
- W. Brent Edwards and Guillaume Y. Millet conceived and designed the experiments, performed the experiments, authored or reviewed drafts of the paper, and approved the final draft.
- Alena M. Grabowski analyzed the data, authored or reviewed drafts of the paper, and approved the final draft.

### Human Ethics

The following information was supplied relating to ethical approvals (i.e., approving body and any reference numbers):

University of Calgary Conjoint Health Research Ethics Board.

### Data Availability

The data and code are available at Github: https://github.com/alcantarar/Recurrent_GRF_Prediction.

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
