# Peer review of "Predicting continuous ground reaction forces from accelerometers during uphill and downhill running: a recurrent neural network solution"

_PeerJ, doi:10.7717/peerj.12752_

## Round 0.1 · original submission · Minor Revisions

Thank for your submission, “Predicting continuous ground reaction forces from accelerometers during uphill and downhill running: A recurrent neural network solution”. We’ve received reports from two reviewers, both of whom agree that only Minor Revisions are required.

Both reviewers primarily raise concerns about the research methods, including the effects of filtering, possible artifacts arising from details of IMU attachment, and other decisions made during data collection or processing. Please take a careful look at these comments and add clarifications or caveats where required.

I had one additional concern not mentioned by the reviewers. You found that the neural network model had trouble reconstructing impact transients observed during the faster trials (leading to greater error in predicting loading rates). The explanation for this result (i.e., lack of input from distal IMUs) makes sense to me. However, I do wonder about the implications of this for the overall utility of the model. Specifically, impact transient magnitudes may be predictive of running-related injury (i.e., Altman, Allison R., and Irene S. Davis (2012). "Barefoot running: biomechanics and implications for running injuries." Current Sports Medicine Reports 11: 244-250.). Given that coaches and athletes may want to use your method and model to gauge injury likelihood during a competitive season, the decreased ability to reconstruct impact transients would seem to be a disadvantage. Please comment on this.

I look forward to receiving your revised manuscript.

Reviewer 1 ·

Basic reporting

No Comment

Experimental design

No Comment

Validity of the findings

This study uses LSTM neural networks to predict GRF characteristics from sacral IMUs and temporal gait characteristics from shoe IMUs. This work is novel and the question is highly relevant for bringing gait analysis out of the lab. The study should be published with minor revisions.

Justification for application to track and trail running is limited not everyone is interested in trail running, however, there is certainly good application for clinical, rehab and geriatric applications. These could be identified to augment the justification

One methodological question. Is it necessary to align IMU and treadmill frames of reference? The neural network can likely learn the patterns regardless of this alignment, but would prediction be better if this step was performed. Certainly in non-lab situations this alignment will not be possible since one does not know substrate orientation. Please discuss this point in the MS.

The authors discuss the methods poor ability to detect the presence of an impact peak? Were results better for heel strike runners vs. midfoot (who normally have much lower impact peaks)? Did filtering at 30Hz or 20Hz remove impact peaks? Please discuss the effects of filtering on this pattern in the paragraph (lines 316-330)

Error metrics (Table 3) show good predictability for most of the metrics with the exception of loading rate. This will be strongly influenced by filtering of impact peaks. Please address this.

To what extent are results influenced by duration of the trials? I would expect that trail running on varied terrain will result in substantially altered GRF for subsequent footfalls. Could you address how this results might differ with more variable input data?

The authors are correct to discuss the effect of attachment methods for the IMUs. In addition there is likely variation in skin movement at different body sites (and among subjects) depending on the quantity and properties of subcutaneous fascia. Some of this can addressed by comparing the vibrational frequencies of the IMUs. See Smeathers JE, (1989). Proceedings of the Institute of Mechanical Engineers, 203: 181-186. Also see Smith http://hdl.handle.net/2436/623290. Please address this variable in the discussion.

Additional comments

Minor point:

Insert 2 commas in the sentence lines 272-5 “..network, given ….sacral acceleration, can predict”

·

Basic reporting

-The document is well-written and easy to follow.
-Figures are well-designed and add to the strength of the manuscript.
-Sample data is not misleading.
-Models and raw data are freely available online via the provided GitHub link, as well as thorough documentation.
-Manuscript is appropriately self-contained. Extensions from original hypothesis (i.e. exploring accuracy of individual models, excluding foot strike info) are very appropriate.

Experimental design

-Manuscript fits within the aims and scope of the journal.
-Research question is well defined, relevant, and meaningful.
-Approach is technically sound and ethical – recommendations for utilizing machine learning for studies of human biomechanics have been followed.
-Methods are described in sufficient detail to replicate.

Validity of the findings

-Manuscript is valuable, as the data are readily available, methodology is clear, and documentation of approach is abundant.
-Data are provided via GitHub link.
-Conclusions are well stated, linked to original research question, and are limited to the supporting results.

Additional comments

While I am not an expert in machine learning approaches, I was able to understand the methods used by this research team and the findings/limitations. Below are a few comments/suggestions/questions that I hope the authors can use to improve the manuscript.

Line 64: The argument is made in the manuscript that other approaches require normalizing waveforms to the duration of the step prevent the calculation of temporal biomechanical variables. However, if the duration is known (and used to perform the normalization), can’t temporal variables be calculated?

Line 89: If GRF is calculated ‘only’ during the stance phase, isn’t that as good as calculating for the entire cycle since GRF = 0 during the flight phase? Not sure I’m missing something here…

Line 121: I was surprised that only bi-axial accelerometers were used – would the authors expect results to change if 3 axes of data were available?

Lines 129-134: Is there any other motivation to filtering cut-off values other than improving the prediction accuracy? Why not the same cut-off for both systems?

Data Processing: Was any attempt made to ensure the accelerometers were mounted the same on everyone (i.e. same sensor to segment alignment)? Was there a chance that sensors were placed slightly out of alignment (if assuming a physical alignment with body structures)? If so, how would that affect the results here?

Data Processing: Overall, there seems to be many choices made in order to get the best results possible (cut-off values, window size, making negative accelerations = 0). Is this common? Is there a systematic way to make these choices?

Discussion: As a reader, I am left wondering what features were most valuable for accurately predicting the nGRF. I would imagine that a simple model with participant mass, height (leg length), running speed, stride frequency, and slope would do pretty well in predicting the nGRF waveform – how much value do the accelerometer features add?

Line 356: The citation of Vitali et al. 2019 is incorrect for the sentence – Vitali and colleagues did not use GPS and tested more than 3 subjects.

Thank you for providing all the data, code, and tutorial – this manuscript motivated me to download and try out Python for the first time.

---

## Round 0.2 · accepted · Accept

The authors have done an excellent job responding to all of the reviews. I think this is an important and useful study, and I look forward to seeing it in print at PeerJ.